# “We Just Don’t Know Where They Are”: The Geographical Distribution of Exercise Classes for Older People, Including Those Living with Dementia in the East Midlands

**DOI:** 10.3390/ijerph20032142

**Published:** 2023-01-24

**Authors:** Annabelle Long, Stephen Timmons, Claudio Di Lorito, Vicky Booth, Pip Logan

**Affiliations:** 1Centre for Rehabilitation and Ageing Research, School of Medicine, University of Nottingham, Nottingham NG7 2UH, UK; 2Nottingham University Business School, University of Nottingham, Nottingham NG8 1BB, UK; 3Nottingham University Hospitals NHS Trust, Queens Medical Centre, Nottingham NG7 2UH, UK; 4Nottingham City Care Partnership, Nottingham NG6 8WR, UK

**Keywords:** exercise classes, dementia, older people, GIS, distribution

## Abstract

Older people living with dementia are advised to exercise to remain independent. Although several exercise classes for older people take place across the UK, there is limited information about the geographical distribution of these classes. This study identified the location and explored the population characteristics of the classes in a UK region, to aid improved access to exercise. Using a geographical information system, data were collected on population characteristics, including size and age, socio-economic status, and rurality of the exercise classes in one area of the UK (East Midlands, population 5 million). The relationship between data sets was explored and a visual representation of these patterns was provided. A systematic internet search identified 520 exercise classes, evenly spread across the region and areas of socio-economic deprivation: 471 (90%) were in urban areas; 428 (80%) were in areas where less than 20% of the population was over 65 years of age; and 13 (2%) stated that they were suitable for people with dementia. People living with dementia are less likely than older people without dementia to have access to exercise classes.

## 1. Introduction

Dementia is an umbrella term which describes a range of cognitive and behavioural symptoms, including memory loss, judgement and changes in personality, that can lead to a substantial decline in cognitive function, and difficulties with everyday activities [1,2,3].

The World Health Organization (WHO) action plan for dementia recognises that the condition is a major cause of disability and dependence in older adults, with 47 million people affected worldwide in 2015 1. In the UK alone, in 2019, it was estimated that 885,000 people were living with dementia [4].

People can still live well with dementia and keeping as active as possible is one of the approaches that can help both with the management of symptoms and sustaining independence.

Exercise has been shown to have many benefits for older people, and several systematic literature reviews have identified that exercise has the potential to improve the physical function and psychological well-being of people living with dementia [5,6,7,8]. However, to maintain its benefit, exercise needs to be sustained over time.

People with dementia are willing and able to attend exercise classes [9] under the right conditions, but there is limited information to inform potential attendees on the current provision of exercise classes.

Geographically mapping existing exercise classes could be an effective first step to investigate current service provision for people living with dementia, by determining the location of classes in comparison to population characteristics such as age.

As well as providing key information on service provision, maps can also reveal patterns and provide answers to more complex questions, such as relationships that may exist between sets of data—for example, the location of exercise classes for older people compared to the concentration of older people in a particular area [10,11]. It can also be a useful tool in healthcare planning, as the accessibility and utilisation of health services can be mapped, alongside planning the location of new services to maximise these factors for the representative population [12].

This type of mapping has been used widely in healthcare, from measuring accessibility to specialist services, describing the geographic patterns of fall injuries among elderly people, mapping the pattern of a disease, and exploring the risk factors of a particular disease [13,14,15,16].

Whilst some international research has previously looked at the distribution of facilities stratified by socioeconomic status [17,18,19], there is very little information available showing data trends from the UK. This study aims to investigate where exercise classes are located in the UK region of the East Midlands and how they are distributed with regards to location, population age, socio-economic status, and population density, to allow the effective planning of the location of classes to enable better access for the population they are designed for.

## 2. Materials and Methods

This study used a geographic information system (GIS), a software system that allows data input, storage, mapping, and spatial analysis. GIS provides a different understanding of these relationships and can also support decision-making, such as where to place a new exercise class [20]. GIS provides a visual representation of these patterns that goes further than a database, as it integrates different sources of data. These data can then be linked by spatially matching datasets in overlays which can be analysed to produce new information [21]. GIS can provide information about the geographical distribution of exercise classes for older people analysed in relation to population distribution, socio-economic status, and population density. This can generate more detailed information about accessibility to classes for the population they are intended to support [20]. The information also enables geocoding of each individual location, to create a point location map [22].

### 2.1. Study Area

The East Midlands is a region of England consisting of five counties: Derbyshire; Leicestershire and Rutland; Lincolnshire; Northamptonshire and Nottinghamshire. It covers an area of 15,627 km^2^, with an approximate population in 2018 of almost 5 million people [23]. The area is diverse in terms of socioeconomic status, with a wide range of deprivation [24] and around 30% of the population living in rural areas [25]. Data from the Office of National Statistics estimated that there were just over 800,000 people over 65 years living in this region in 2019 [26]. Figures from Alzheimer’s Research UK in 2015 suggested that, at that time, there were just over 68,000 people living with dementia across the region [27]. Data from the UK government shows that dementia prevalence in the East Midlands is 0.9%, which is higher than the national average of 0.8%, and ranges from 0.8% in Northamptonshire and Leicestershire to 1.0% in Lincolnshire [28].

### 2.2. Study Data

The following data were collected:Demographic data: Percentages of population over 65 years were gathered from the 2011 census data, which was available through the ArcGIS software [29].Socio-economic data: UK Index of Multiple Deprivation 2015 was gathered from data available through the ArcGIS software [30].Population density data: World Population Density Estimate 2016 was gathered from data available through the ArcGIS software [31].Location data: Postcode data for exercise classes were gathered through an internet search using Google.

Data were also gathered from the Office of National Statistics from mid-2019 on the number of people over 65 years of age living in each county in the East Midlands [26] and, from Alzheimer’s Research UK, on the number of people living with dementia in each Clinical Commissioning Group (CCG) within the East Midlands in 2015 [27].

### 2.3. Data Collection

An internet search using a Google search was undertaken in February 2020. Location data were gathered for exercises classes for older people in each county within the East Midlands region. The search terms “exercise” AND “older people” were added to a county location e.g., AND “Derbyshire”.

Searches indicated several national chains of exercise providers, which offered classes in each county, alongside specific individuals or local groups in each area. A spreadsheet was created to gather basic information about each company, group or individual that was identified through this search. The larger national chains of exercise providers offered classes across the whole country but used a class finder service on their website, which could be used to locate classes in each separate county of the East Midlands.

A scaled map was created using Google Maps for each county that forms the East Midlands region. These maps were then merged to form a visual boundary map of the region, which was used to establish a centre point for each county. This centre point enabled an appropriate postcode and radius to be calculated, which was used to search and gather details of all the classes in each county.

Separate sheets were created within the spreadsheet for each county to record details of each class, including the organisation running the class; type of class (including if they stated the class was appropriate or specific for people living with dementia); age range (if stated); and the location, postcode, and contact details of the instructor.

To ensure that classes were not missed, a centre point of the East Midlands region was also established, alongside an appropriate postcode and radius. This was then used to check the entire region in a ‘second sweep’, ensuring no class falling within the boundary was missed.

### 2.4. Geocoding and Mapping

The information gathered from the internet searches was geocoded, using the postcodes for each individual class location, to enable these locations to be mapped. A geodatabase for each individual county was created in ArcGIS Pro. These geocoded point locations were then used in conjunction with an Ordnance Survey background map, to show the distribution of exercise classes across the East Midlands region [32].

Further data were gathered through the ArcGIS software regarding the percentages of people over 65 years, levels of socio-economic deprivation and population density levels. These data were overlayed with the point location maps, to provide additional information about the distribution of classes in relation to age, socio-economic status, and population density.

The data gathered from the Office of National Statistics and from Alzheimer’s Research UK were used to calculate population to available class ratios [33,34].

### 2.5. Spatial Analysis

Spatial analysis tools within the ArcGIS software were used to calculate the number and percentage of classes within each county with respect to population age, socio-economic status, and population density.

Demographic data gathered from the 2011 census were shown with respect to the percentage of population over 65 years of age within each geographical location (area data) [29], which were represented by polygons displayed on a background location map.

Socio-economic data gathered from the United Kingdom Index of Multiple Deprivation from 2015 [30] were again shown with respect to each geographical location, which were also represented by polygons displayed on a background map.

Population density data gathered from the World Population Density Estimate from 2016 [31] was only available in a raster data format, which could not be used to further analyse in conjunction with class location data. These data required conversion to polygon data to enable further analysis, which was completed with the Raster to Polygon conversion tool within ArcGIS Pro. Once data were converted, they could be displayed on a background map.

Using the analysis tool available within the ArcGIS software, the location of classes (point data) was overlayed on each set of available polygon (areal) data (age of population, deprivation index and population density) using the spatial join tool. This enabled a calculation of the number of points present within each polygon. These datasets were then used to calculate the percentage of classes present in areas split by age of population, level of deprivation and level of urbanisation.

## 3. Results

### 3.1. Location

The population over 65 years of age in the East Midlands was just over 800,000, with just over 68,000 people diagnosed as living with dementia. The internet search found 520 exercise classes advertised as being appropriate for older people in the East Midlands region, but these were not evenly spread either by county or by population numbers. Figure 1 below shows the point location map of all the exercise classes for older people in the East Midlands. Classes specifically advertised as being dementia-friendly were located using a red pin, whilst other classes for older adults were located using a blue pin.

Of the 520 classes that were located, only 13 (2%) were specifically advertised as being suitable for people living with dementia. Table 1 shows the number of exercise classes separated by county, alongside the number of people over 65 years living in each county [26] compared to the number of classes for people living with dementia and the approximate figures for each county from 2015 [27].

### 3.2. Classes Compared to Population over 65 Years

Figure 2 shows the location of exercise classes in the East Midlands, set against the percentage of the population over 65 years. Raw data has been provided in Table A1 in Appendix A.

There were 201 exercise classes for older people in Derbyshire, which were spread quite evenly. There were four dementia-specific exercise classes: one in the city of Derby; one in the very south of the county; and two in mid Derbyshire. One-hundred-and-fifty-six (78%) classes were in areas where less than 20% of the population was over 65 years, with 45 (22%) in areas where there was a higher proportion of people over 65 years.

There were 82 exercise classes for older people in Leicestershire, which were spread across the county, with a cluster of 14 classes in the city of Leicester and its surrounding areas. There were three dementia-specific exercise classes: two in the north of the county and one in the east. Nineteen (23%) classes were in areas where less than 15% of the population was over 65 years, with a further 58 (71%) in areas where between 15.1% and 19.9% of the population was over 65 years. Just five (6%) classes were in areas where more than 20% of the population was over 65 years.

There were 91 exercise classes for older people in Lincolnshire, which were spread widely throughout the county. There was a cluster of classes in the city of Lincoln and its surrounding area. There were five dementia-specific classes in Lincolnshire: one in Lincoln and two in its surrounding villages. However, there was also one in the south of the county and one in the east. Lincolnshire had the most dementia-specific classes in the East Midlands. Twenty-one (23%) classes were in areas where less than 15% of the population were over 65 years, and these were all within Lincoln and its surrounding area. Fifty-three (58%) classes were in areas where over 20% of the population was over 65 years, with 15 (16%) found in an area where nearly 30% of the population was over 65 years.

There were 79 exercise classes for older people in Northamptonshire. These were mainly clustered around the towns of Northampton, Kettering, Wellingborough, and Corby. There were no dementia-specific exercise classes in Northamptonshire. There were no classes in Northamptonshire located in areas where there was a high proportion of the representative population with all classes found in areas where less than 20% of the population was over 65 years. Thirty-five (44%) classes were situated in areas where less than 15% of the population was over 65 years, with 44 (56%) found in areas with between 15.1 and 19.9% of the population was over 65 years.

There were 67 exercises classes for older people in Nottinghamshire. However, classes were largely clustered in Nottingham city and its surrounding area. There was one dementia-specific class, which was in Nottingham. There were no classes in Nottinghamshire, which were in areas where there was a high proportion of the representative population, with all classes found in areas where less than 20% of the population was over 65 years. However, in contrast to Northamptonshire, only 12 (18%) classes were found in areas where less than 15% of the population was over 65 years, with 55 (82%) situated in areas where there was between 17.1–19.9% of the population over 65 years.

### 3.3. Classes Compared to Levels of Deprivation

Figure 3 shows the location of exercise classes in the East Midlands compared to the Index of Multiple Deprivation. This index ranks small areas in the UK from the most deprived (1) to least deprived (32,844). The map below shows the deprivation deciles which describe the percentile that the area falls within, e.g., 1 being the most deprived 10% and 10 being the least deprived 10%. Raw data has been provided in Table A2 in Appendix A.

Derbyshire had a mixture of areas of both high deprivation (1.0) and low deprivation (10), and classes were well distributed in all areas, ranging from nine classes in the highest areas of deprivation to 15 classes in the lowest areas. Overall, 94 (45%) classes were in areas which were rated from 1–5 on the Index of Multiple Deprivation, with 115 (55%) located in areas rated from 6–10.

Leicestershire also had a mixture of areas of both high deprivation (1.0) and low deprivation (10), and classes were well distributed in all areas. However, in contrast to Derbyshire, 52 (63%) classes were in areas which were rated from 1–5 on the Index of Multiple Deprivation, with just 30 (37%) located in areas rated from 6–10.

Lincolnshire had an even distribution of classes across its areas of high (1.0) and low deprivation (10), ranging from eight classes in the highest areas of deprivation to five classes in the lowest areas. Overall, 49 (54%) classes were in areas which were rated from 1–5 on the Index of Multiple Deprivation, with 42 (46%) located in areas rated from 6–10.

Northamptonshire also had an even distribution of classes across its areas of high (1.0) and low deprivation (10). However, in a similar pattern to Derbyshire, there were less classes in areas of high deprivation, with 36 (46%) classes located in areas which were rated from 1–5 on the Index of Multiple Deprivation and 43 (54%) located in areas rated from 6–10.

Nottinghamshire had a mixture of areas of both high deprivation (1.0) and low deprivation (10), and classes were reasonably well distributed in all areas. However, in contrast to an area like Leicestershire, only 26 (39%) classes were in areas which were rated from 1–5 on the Index of Multiple Deprivation, with 41 (61%) located in areas rated from 6–10.

### 3.4. Classes Compared to Population Density

Figure 4 shows the location of exercise classes in the East Midlands compared to population density. Raw data has been provided in Table A3 in Appendix A.

Exercise classes in Derbyshire were mainly located in urban areas, with 64 (32%) classes found in light urban and 125 (62%) in urban areas. Twelve (6%) classes were located in settled areas, some of which were within the Peak District National Park. There was a tendency for classes to be located along major roads within the county.

Exercise classes in Leicestershire were predominately found in urban environments, with 12 (15%) located in light urban areas and 70 (85%) in urban areas. There were no classes found in rural or settled areas. Classes in Leicestershire did not have the same location pattern along major roads that was seen in Derbyshire.

Despite its more rural nature, classes in Lincolnshire were also located mainly in urban environments, with 28 (31%) in light urban and 59 (65%) in urban areas. With four (4%) classes located in settled areas and none in rural settings, the distribution was very similar to that seen in Derbyshire.

Classes in Northamptonshire were again mainly clustered around light urban (21; 27%) and urban areas (57; 72%), with just one (1%) class found in settled areas and none in rural areas. In Northamptonshire, like Derbyshire, there was a tendency for the classes to be located along major roads within the county.

Nottinghamshire had the most even spread of classes, with seven (10%) classes found in settled areas. Only 35 (51%) classes were in urban areas, which was the lowest percentage of all five counties, with 27 (39%) found in light urban areas.

## 4. Discussion

### 4.1. Summary of Findings

This study was the first of its kind to systematically map, in detail, exercise classes appropriate for people with dementia, with the aim of improving accessibility. Five-hundred-and-twenty exercise classes dedicated to older people were located across the East Midlands region, but only 13 (2%) were advertised as appropriate for people with dementia, meaning there were limited opportunities for this cohort to take part. Putting this into context, there were over 68,000 people with dementia living in this area in 2019; if all wanted to attend, each class would have over 5,000 attendees and, dependent on their location, they would have to make a potential round-trip of between 70 km to 167 km. To make matters worse, many of the classes were held in locations where a low percentage of the population were over 65 years, limiting their accessibility to many older people. Most classes (67%) were in urban areas, meaning older people in rural areas were disadvantaged. Some classes were located along major roads, which may have increased accessibility for car users but may have deterred public transport users and pedestrians, and classes were located evenly between areas of low (51%) and high (49%) socioeconomic status, which will have encouraged use.

### 4.2. Comparison with Other Studies

There are no studies that have investigated the distribution and use of exercise facilities for people with dementia with which to compare our findings. The following studies have explored other populations and different services, but their findings need to be kept in context. Older people living with dementia experience additional barriers, including reliance on others for access, reduced confidence around people and difficulties with motivation [9], which may not be considered in more general population studies.

Pedersen et al. [15] reported that a reliance on public transport led to longer travel times to attend cardiac rehabilitation classes in Denmark, which became a barrier to attendance. In our study, classes in both Derbyshire and Northamptonshire were located on major roads, which may have favored car owners, but we do not know for certain. Lincolnshire, in comparison, has classes that are spread across the county in smaller towns and villages, which may increase access for pedestrians and those using public transport. Access to private transport decreases in older populations [35], leaving people reliant on public transport or walking, so we would recommend exercise classes consider transportation.

Higgs et al. [36] studied accessibility, for a general population, to a wide range of sporting facilities in Wales and found use was greater in more socially deprived areas and lower in the more affluent areas, which was similar to the present study that found a higher percentage of classes (63%) in more socially deprived areas. This was in contrast to Nottinghamshire, where a higher percentage of classes were located in more affluent areas, which was similarly found in a study by Christie et al. [37], who looked at the accessibility of renal replacement therapy units in South and Mid Wales.

Yin [38] investigated differences in accessibility to pre-natal care in Georgia (USA) and found that access to pre-natal care decreases as rurality increases, which is similar to our finding that the majority (96%) of classes in the region were located in light urban or urban areas.

### 4.3. Strengths and Limitations

A strength is that it is the first study that explored the distribution of exercise classes advertised on the internet and compared them with population variables, allowing future comparison studies. It was completed by applying a systematic approach to the internet search, using scaled maps and rigorous geolocation. With this level of detail, we are confident that no exercises advertised on the internet were missed; however, we must acknowledge that classes with no online presence would not have been included.

Although it was noted that classes in Derbyshire and Northamptonshire were located on major roads, a further limitation was that accessibility, in terms of distance or travel times to a class based on access to a car or use of public transport, was not included in this study.

A further limitation is that this study was completed in 2020, just before the COVID-19 pandemic, which led to a lockdown in the UK. It has not been repeated to check how many classes still exist. However, the results are still relevant, as they provide a baseline of dementia services at that time.

### 4.4. Further Research and Recommendations

#### 4.4.1. Practice Recommendations

The lack of dementia-specific exercise facilities exposed by this research is not acceptable when we know that exercise is recommended in all clinical guidelines to prevent falls, reduce immobility, and help people with dementia remain independent. We recommend that organisations should consider investing in training a greater number of exercise instructors to deliver exercise classes that are appropriate for people living with dementia, spread across rural and urban locations, with accessible public transport, parking, and are offered in different languages to enable equitable access.

#### 4.4.2. Research Recommendations

We recommend that any new services are studied to make sure they are meeting the needs of people with dementia and are cost-effective. We recommend that researchers and policy makers should consider the GIS as a tool which when planning future public health decisions, as it can highlight inequalities across seldom-heard communities. Considering additional barriers that may exist for these communities may allow facilities to be more inclusive for the communities they serve.

## 5. Conclusions

This study located and mapped 520 exercise classes for older people in the East Midlands. Classes are not uniformly distributed across the region, and there are very few exercise classes which state they are appropriate for people with dementia. Most classes are in urban areas with limited coverage in rural areas. Many classes are in areas where less than 20% of the population are over 65 years of age. However, classes are spread evenly across the region between areas of differing socio- economic status. Understanding where exercise classes are currently located with reference to these variables, especially population age, may be useful for policymakers when looking to locate new classes to increase accessibility for the population for whom they are designed. Future research could explore the accessibility in terms of distance or travel times to an exercise class, which was not looked at as part of this study. A spatial analysis looking at distances and travel times to exercise classes, based on both access to a car and use of public transport, could provide additional information for planners or policy makers when looking to locate new services.

## Figures and Tables

**Figure 1 ijerph-20-02142-f001:**
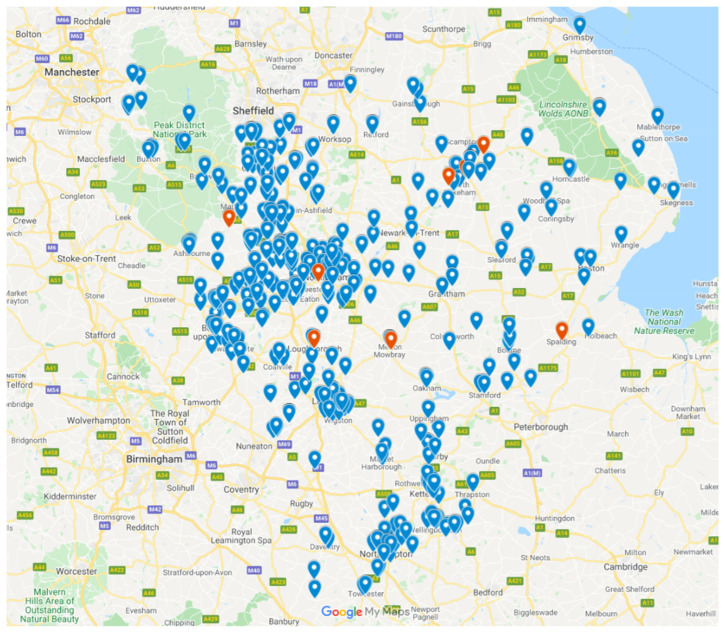
Point location map of classes in the East Midlands.

**Figure 2 ijerph-20-02142-f002:**
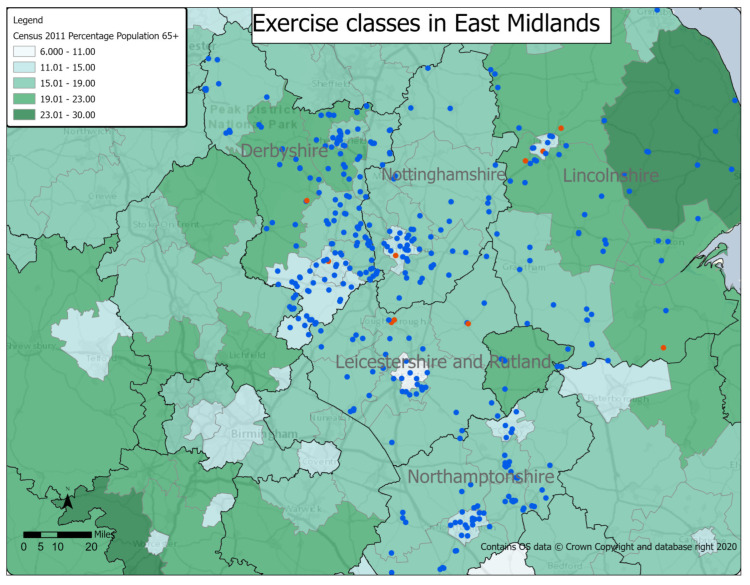
Location of exercise classes in East Midlands, overlaid with the percentage of the population over 65 years.

**Figure 3 ijerph-20-02142-f003:**
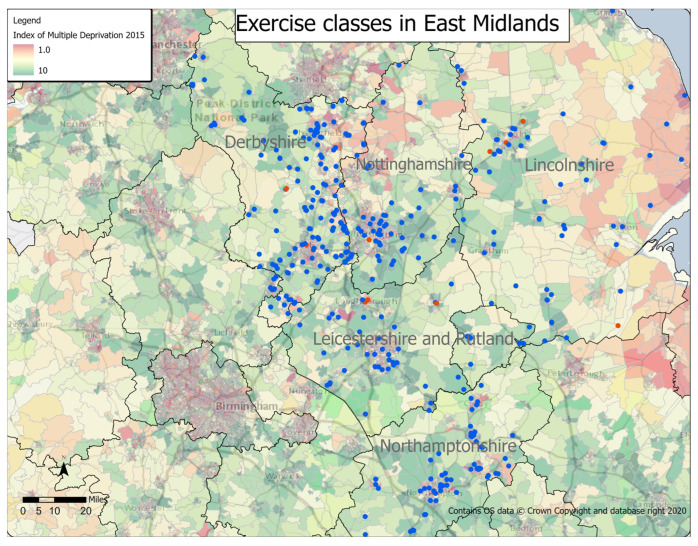
Location of exercise classes in East Midlands, overlaid with the Index of Multiple Deprivation.

**Figure 4 ijerph-20-02142-f004:**
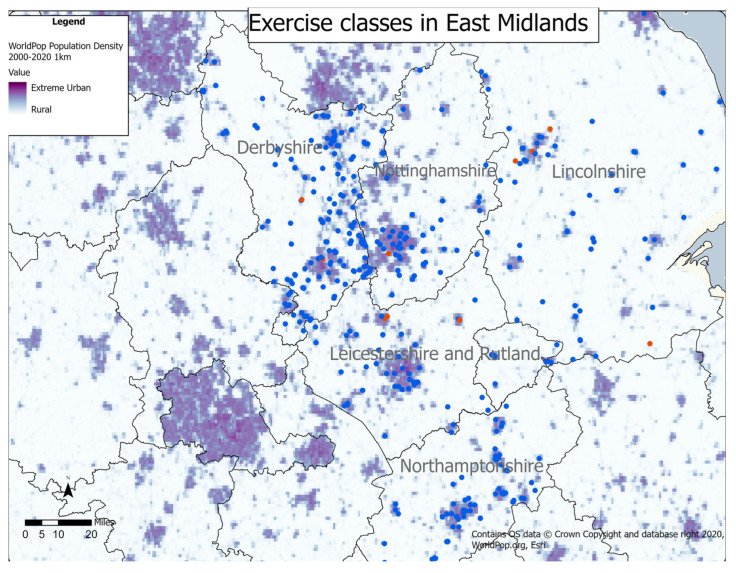
Location of exercise classes in East Midlands, overlaid with population density estimates.

**Table 1 ijerph-20-02142-t001:** Number of classes and population over 65 years.

County	Number of Classes for Older People (People with Dementia)	Population Over 65 Years (Population Over 65 Years with Dementia)	Ratio of Over 65s/Class (People with Dementia/Class)
Derbyshire	201 (4)	174,956 (14,749)	870 (3690)
Leicestershire	82 (3)	144,892 (13,723)	1767 (4574)
Lincolnshire	91 (5)	179,805 (17,596)	1976 (3519)
Northamptonshire	79 (0)	136,682 (8883)	1730 (-)
Nottinghamshire	67 (1)	173,311 (13,245)	2586 (13,245)

## Data Availability

Publicly available datasets were analyzed in this study. This data can be accessed through links provided in the references. Other data presented in this study is available in the Appendix A.

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
