# Peer review of "“We Just Don’t Know Where They Are”: The Geographical Distribution of Exercise Classes for Older People, Including Those Living with Dementia in the East Midlands"

_ijerph, 2023, doi:10.3390/ijerph20032142_

Round 1

Reviewer 1 Report

This study showed a basic data for the geographical distribution of exercise classes. However, I have some comment below.

1. In figure 1 to figure 4, where is the five counties located? Please show the boundaries and five counties’ name in the map.

2. Socioeconomic status is different in countries and area deprivation index is affected by the socioeconomic status in each country. This study showed that the geographic information and the number of older adults or people living with dementia, however, did not show the causality of geographic distribution and health for older adults. I think this article contains some exaggerated phrases like following. Please reconsider the wording.

Line 300303: “enables comparisons to be drawn with classes located over the worldwide” or Line 379381 “Understanding where exercise classes are currently located with reference to these variables can be useful for policymakers enabling them to effectively plan where to locate any new classes to increase accessibility for the population for whom they are designed.”

3.  In discussion, the comparison of previous study and limitation of this study were mixed. Additionally, the discussion was redundant. Please reconsider the structure of discussion.

4. The location of exercise classes is not solely determined by the geographic distribution. This study was not analyzed with considering the public transportation or the proportion of using their own cars. I think this is the limitation of this study.

Author Response

Responses to Reviewer 1

Many thanks for taking the time to read and review our article. We have considered your comments and have provided the following responses.

Comment 1: In figure 1 to figure 4, where is the five counties located? Please show the boundaries and five counties’ name in the map.

Response 1: Thank you for your comment which was very helpful. I know the area well and didn’t appreciate the benefit of having the counties lines on the map. Unfortunately Google does not allow you to add county lines but I have been able to amend figures 2 – 4 to include the boundaries and names of the counties.

Comment 2: Socioeconomic status is different in countries and area deprivation index is affected by the socioeconomic status in each country. This study showed that the geographic information and the number of older adults or people living with dementia, however, did not show the causality of geographic distribution and health for older adults. I think this article contains some exaggerated phrases like following. Please reconsider the wording.

Line 300-303: “enables comparisons to be drawn with classes located over the worldwide” or Line 379-381 “Understanding where exercise classes are currently located with reference to these variables can be useful for policymakers enabling them to effectively plan where to locate any new classes to increase accessibility for the population for whom they are designed.”

Response 2: Thank you for your comment. I appreciate the wording was slightly over exaggerated and so have amended the phrasing so that causality is not implied.

Comment 3: In discussion, the comparison of previous study and limitation of this study were mixed. Additionally, the discussion was redundant. Please reconsider the structure of discussion.

Response 3: Thank you for your comment. On re-reading the article, I appreciated that there was a lot of repetition in the discussion, and it was not structured as effectively as it could have been. This has now been restructured and rewritten. I have added sub headings to the discussion but they are not necessarily needed if the journal structure is not to have sub-headings.

Comment 4: The location of exercise classes is not solely determined by the geographic distribution. This study was not analyzed with considering the public transportation or the proportion of using their own cars. I think this is the limitation of this study.

Response 4: Thank you for your comment. The conclusion does mention that a spatial analysis looking at travel times based on access to a car and use of public transport but as suggested this has not been included as a limitation. I have now added this as a limitation to the study as well as in the conclusion.

Reviewer 2 Report

Review report

Article title:  

“We Just Don’t Know Where They Are” – The Geographical Distribution of Exercise Classes for Older People Including Those Living with Dementia in the East Midlands

Summary

The authors present a research article where they explore the geographical distribution of exercise classes for older adults and people living with dementia in the East Midlands. The study was conducted using an internet search and data was treated using an appropriate software.

General questions:

·          Is the manuscript clear, relevant for the field and presented in a well-structured manner?

Yes, the manuscript is relevant for the field, well written and structured.

·          Are the cited references mostly recent publications (within the last 5 years) and relevant? Does it include an excessive number of self-citations?

25 out of 44 references are older than 5 years; nevertheless, the references used are appropriate and relevant. Many references are from internet sources and not peer reviewed sources.  

·          Is the manuscript scientifically sound and is the experimental design appropriate to test the hypothesis?

The aims of the manuscript are to locate exercise classes for older individuals and people with dementia and to correlate the location with several indicators of population distribution. The experimental approach is adequate to the objectives of the study.

·          Are the manuscript’s results reproducible based on the details given in the methods section?

The details given in the methods section are sufficient to reproduce the research that is presented.

·          Are the figures/tables/images/schemes appropriate? Do they properly show the data? Are they easy to interpret and understand? Is the data interpreted appropriately and consistently throughout the manuscript?

Figures are good and tables are also fine.

·          Are the conclusions consistent with the evidence and arguments presented?

Yes

·          Are the ethics statements and data availability statements adequate?

No ethics statement is presented. The data availability statement is provided and adequate.

Article General concept comments:

In overall this is a work supported by an interesting objective. It is innovative and refreshing.  

Specific comments

I have only few questions regarding this work.

1-     How confident are the authors that the internet search to locate the exercise classes retrieved accurate results.

2-     Would it be interesting also to differentiate classes that are provided by the private sector (for instance chains of exercise providers) and classes provided by community centers (social centers)?

3-      While many of the data used in this work is easy to understand, the Index of multiple deprivation is not. Could the authors provide a brief description of the index?

Author Response

Responses to Reviewer 2

Many thanks for taking the time to read and review our article. We have considered your comments and have provided the following responses.

Comment in Report: 25 out of 44 references are older than 5 years; nevertheless, the references used are appropriate and relevant. Many references are from internet sources and not peer reviewed sources.  

Response: Thank you for highlighting this. 11 of the references refer to the data that has been used in the study such as the figures gathered for number of people over 65 in each county and so are gathered from publicly available sources such as Ordnance Survey and Office for National Statistics and so not peer reviewed

Comment 1:  How confident are the authors that the internet search to locate the exercise classes retrieved accurate results.

Response 1: Thank you for this comment. We are fairly confident that the search to locate the classes retrieved accurate results. The search was completed systematically with the use of boundary maps and postcodes to try to ensure that classes were not missed. There was a chance that after the search was completed that classes closed - especially following COVID - and classes could have been missed if they had no internet presence at all. However, when geocoding the postcodes for each location for the mapping purposes there was evidence on the internet that classes were taking place at the locations mapped for example a church hall that was advertised as having a class in a class finder, located through the postcode also indicated that the class was running.

Comment 2: Would it be interesting also to differentiate classes that are provided by the private sector (for instance chains of exercise providers) and classes provided by community centers (social centers)?

Response 2: Thank you for this suggestion. It would definitely have been interesting to differentiate the providers as well, but this was difficult to find out from the internet alone. There was very little information available to show if these were private or community run and what information there was indicated that the vast majority were private. A survey which was completed alongside the mapping for my PhD had a very limited response (likely due to COVID). This did look to differentiate the type of provider as this was to be part of purposive sampling for a later part of my PhD. Unfortunately, due to the limited response this was not possible.

Comment 3: While many of the data used in this work is easy to understand, the Index of multiple deprivation is not. Could the authors provide a brief description of the index?

Response 3: The Index of Multiple Deprivation was a data map provided through ESRI from the Ordnance Survey. This index ranks every small area in England from 1 (most deprived area) to 32,844 (least deprived area). This map shows the deprivation deciles which describes the percentile that the area falls in e.g 1 = the most deprived 10% and 10 being the least deprived 10%. I hope that this explains the index and has been added to the article.

Round 2

Reviewer 1 Report

Dear authors,

I think your article was refined precisely according to our comments.